# A New Face of the Old Gene: Deletion of the *PssA*, Encoding Monotopic Inner Membrane Phosphoglycosyl Transferase in *Rhizobium leguminosarum*, Leads to Diverse Phenotypes That Could Be Attributable to Downstream Effects of the Lack of Exopolysaccharide

**DOI:** 10.3390/ijms24021035

**Published:** 2023-01-05

**Authors:** Małgorzata Marczak, Kamil Żebracki, Piotr Koper, Aleksandra Horbowicz, Magdalena Wójcik, Andrzej Mazur

**Affiliations:** Department of Genetics and Microbiology, Institute of Biological Sciences, Maria Curie-Skłodowska University, Akademicka 19 St., 20-033 Lublin, Poland

**Keywords:** exopolysaccharide, glycosyltransferase, phosphoglycosyl transferase, *Rhizobium*, symbiosis, RNAseq

## Abstract

The biosynthesis of subunits of rhizobial exopolysaccharides is dependent on glycosyltransferases, which are usually encoded by large gene clusters. PssA is a member of a large family of phosphoglycosyl transferases catalyzing the transfer of a phosphosugar moiety to polyprenol phosphate; thus, it can be considered as priming glycosyltransferase commencing synthesis of the EPS repeating units in *Rhizobium leguminosarum*. The comprehensive analysis of PssA protein features performed in this work confirmed its specificity for UDP-glucose and provided evidence that PssA is a monotopic inner membrane protein with a reentrant membrane helix rather than a transmembrane segment. The bacterial two-hybrid system screening revealed interactions of PssA with some GTs involved in the EPS octasaccharide synthesis. The distribution of differentially expressed genes in the transcriptome of the Δ*pssA* mutant into various functional categories indicated complexity of cell response to the deletion, which can mostly be attributed to the lack of exopolysaccharide and downstream effects caused by such deficiency. The block in the EPS biosynthesis at the *pssA* step, potentially leading to an increased pool of UDP-glucose, is likely to be filtered through to other pathways, and thus the absence of EPS may indirectly affect the expression of proteins involved in these pathways.

## 1. Introduction

Bacteria synthesize a range of polysaccharides with a variety of biological functions. Exopolysaccharides (EPS) are structurally and compositionally diverse extracellular carbohydrate polymers secreted and accumulated outside the cells of various microorganisms. EPS is of special importance for nitrogen-fixing rhizobia establishing symbiosis with legume plants. Besides the roles attributed to EPS in the case of other bacteria, i.e., protective properties (e.g., against osmotic stress, temperature, pH, heavy metals, and desiccation), cell adherence to surfaces and biofilm formation, and nutrient uptake [1,2,3,4,5,6], rhizobial exopolysaccharide is also indispensable for effective infection and nodulation of many host plants [7]; its signaling function in this process has been confirmed [8].

The huge structural diversity of EPSs is reflected in the complexity of their biosynthetic pathways involving numerous enzymes engaged in the biosynthesis of both EPS and other glycoconjugates (e.g., lipopolysaccharide, LPS). Genes encoding proteins involved in EPS synthesis in rhizobia are usually clustered on the chromosome or large plasmids [7,9]. The structure of the octasaccharide subunit of EPS in *Rhizobium leguminosarum* bv. *trifolii,* i.e., a symbiont of clover, and a chromosomal cluster (named Pss-I, polysaccharide synthesis) responsible for its biosynthesis have been recognized [10,11,12,13,14]. Pss-I contains an almost entire set of genes encoding putative glycosyltransferases (GTs) responsible for EPS subunit synthesis as well as genes coding for enzymes related to polysaccharide polymerization, chain length control, and export [7,11]. However, the priming glucosyl-IP-transferase PssA, which initiates the synthesis of EPS repeating units by transfer of phosphoglucose to isoprenyl phosphate located in the inner membrane [15], is encoded beyond Pss-I.

Based on deduced amino acid sequences, PssA is a member of a family of UDP-glucose:polyprenyl-phosphate transferases belonging to a large class of phosphoglycosyl transferases (PGTs), which initiate the biosynthesis of various glycoconjugates. PGTs catalyze the transfer of a phosphosugar from a nucleoside diphosphate sugar (NDP-sugar) to polyprenol-phosphate, thus forming a membrane-associated polyprenol diphosphosugar intermediate [16]. Data obtained by Bossio et al. [17] provided evidence that intermediates for the biosynthesis of the octasaccharide repeating unit of the *R. leguminosarum* bv. *trifolii* exopolysaccharide were assembled on a lipid acceptor, and the glucose-1-phosphate was attached to the carrier first. The sequence similarity between PssA of *R. leguminosarum* and SpsB of *Sphingomonas* strain S88 as well as complementation analysis allowed Pollock et al. [15] to conclude that PssA must be responsible for adding the first glucose to isoprenyl phosphate during EPS biosynthesis; however, no direct proof of such PssA activity and specificity towards glucose was provided.

PssA was predicted to be an integral inner membrane protein [18] synthesized in two hypothetical translation variants of 263 and 200 aa [19,20]. PGTs in general are integral membrane proteins but exhibit diverse membrane topology and mechanisms of action [16]. One of the superfamilies within the PTG class are polytopic proteins with 10–11 transmembrane helices (TMHs). Bacterial members of this group are WecA-family proteins which transfer GlcNAc-1-phosphate from UDP-GlcNAc to polyprenyl-phosphate in O-antigen and ECA (enterobacterial common antigen) biosynthesis pathways [21]. The other superfamily is represented by polytopic and monotopic proteins sharing a common core domain homologous to the *Campylobacteri jejuni* PglC protein. It is an essential glycosyl-1-phosphate transferase using UDP-bacillosamine to produce undecaprenyl pyrophosphate bacillosamine, the first step in N-linked protein glycosylation [22]. PglC is a prototypic protein containing the so-called reentrant membrane helix (RMH). It comprises a membrane-inserted domain present in the inner leaflet of the membrane rather than spanning it in a manner typical for bitopic proteins. Structural, biochemical, and computational analyses of PglC and its homologs led to establishment of principles governing reentrant topology determination [23].

It was suggested that the *pssA* gene could be expressed and regulated from three different promoters [24]. Indeed, EPS biosynthesis in *R. leguminosarum* and other rhizobia was shown to undergo complex hierarchical regulation, employing numerous regulatory proteins (PsiA, PsrA, ExoR, RosR) and being influenced by several environmental factors (phosphate and nitrogen limitation, carbon source, legume root exudates) [25]. It was shown that, like other genes encoding priming GTs, the *pssA* gene may be a target for such regulation [26]. Analysis of a 767-bp-long *pssA* upstream region in *R. leguminosarum* bv. *trifolii* 24.1 revealed that the transcription of this gene may be driven by two promoters. The RosR regulator was proved to have a positive effect on *pssA* expression by binding to the RosR-box located in the *pssA* promoter. Phosphate and carbon sources strongly affected *pssA* transcription with putative engagement of PhoB and cyclic AMP (cAMP) receptor protein (CRP-like) [26].

Numerous mutants described for *pssA* (mainly containing transposon insertions) did not produce EPS and showed impaired development of nitrogen-fixing nodules on appropriate host plants as well as pleiotropic effects in rhizobial cells [19,20,27,28]. Similarly, Tn5 insertions in the *pssA* promoter region totally abolished production of EPS in *R. leguminosarum* bv. *trifolii* [26]. The pleiotropic effects described for the *pssA* mutation were also commonly observed in other mutants of *pss* genes and comprised disturbances in symbiotic and physiological traits, such as changes in growth kinetics and motility, alteration in cell-surface properties, and biofilm formation [29,30]. The proteomic approach applied by Guerrreiro et al. [27] revealed that genetic perturbations in the EPS biosynthesis pathway in two *pssA* mutant strains resulted in altered protein profiles in *Rhizobium leguminosarum* bvs. *viciae* and *trifolii*. A set of minimum 22 protein differences were observed. However, among proteins belonging to the postulated *pssA* “stimulon”, only two proteins were then functionally annotated as sharing homology with glycosyltransferases from other gram-negative bacteria (MigA from *Pseudomonas aeruginosa*) and the glutamine-binding periplasmic protein (GlnH from *E. coli*), respectively. Other proteins remained unidentified due to the lack of homologues in the sequence databases available at the time [27]. It was also shown that the extracellular protein profile of the *R. leguminosarum* bv. *trifolii* Rt24.2 *pssA* mutant differed from that of the wild-type bacteria. The mutant secreted higher amounts of some proteins, whereas others were absent [31]. Altogether, these data illustrate the degree to which a mutation of a single gene can affect downstream processes of protein function and regulation, suggesting that enzymatic pathways and regulatory networks are more interconnected and more sensitive to structural changes in the cell than is often thought [27].

The engagement of PssA in such complex networks was frequently postulated. Given the development of new molecular analysis tools, the increase in the amount of well annotated genomic data and multiple structural and functional data on GTs reported during recent years, the knowledge of PssA needs to be updated and extended with the concept of its function, structure, activity, and role in exopolysaccharide and other biosynthetic pathways. To address this issue, we performed a comprehensive analysis of the *pssA* gene function and PssA protein features. Taking advantage of the substantial novelty of the database content related to protein structures, we proposed a new model of the PssA secondary structure and topology and verified it experimentally. Using recombinant PssA, we confirmed its substrate specificity. The *cre-loxP*-based gene replacement system facilitated a precise *pssA* deletion and the Δ*pssA* mutant was analyzed with respect to various phenotypic traits and its transcriptional profile by means of RNA-seq based analysis. 

## 2. Results and Discussion

### 2.1. Deletion of PssA Abolishes EPS Biosynthesis and Affects Other Phenotypes of Rhizobium

To date, several mutants abolishing the function of the *pssA* gene have been described for *R. leguminosarum*. The mutants carried mainly a Tn insertion or antibiotic resistance cassettes upstream or inside the *pssA* coding sequence [19,27,28,31], and the phenotypes of these mutants were described. However, for the purpose of this study, we constructed a markerless *pssA* deletion mutant using the pCM351 allelic exchange vector [32] according to the previously implemented method [33,34]. The constructed deletion comprised the entire coding sequence of *pssA* (Appendix A). The extended upstream region of *pssA* containing promoters and putative binding sites for transcriptional regulators described for this gene [26] were not affected, which was confirmed by DNA sequencing.

The macroscopic phenotype of the Δ*pssA* mutant grown on the standard 79CA medium revealed formation of characteristic small non-mucoid colonies differing significantly from those formed by the wild-type (WT) strain (Figure 1A). The quantitative EPS assay indicated that the Δ*pssA* mutant did not produce any amounts of this surface polysaccharide. The EPS production was rescued in the complemented Δ*pssA*(*pssA*) strain (Figure 1B). No concurrent changes in the LPS profiles in the RtTA1 derivatives were found (Figure 1C).

The Δ*pssA* mutant strain was characterized by minor changes in sensitivity to stress conditions, in contrast to previously described mutants possessing a Tn-5 insertion within *pssA.* They displayed severe pleiotropic effects such as decreased utilization of some nutrients, decreased sensitivity to antibiotics, and increased sensitivity to osmolytes [31]. In this study, from among the tested stressors, only deoxycholate (DOC) significantly affected Δ*pssA* growth, and the mutant was slightly more resistant to the increasing concentration of NaCl (up to 75 mM) (Appendix A). SDS, ethanol, and different pH values of the medium did not have any effect on the mutant growth. The symbiotic properties of the Δ*pssA* strain were significantly affected, like previously published *pssA* mutants of *R. leguminosarum* bvs. *trifolii* and *viciae* [19,20,31]. RtTA1 Δ*pssA* elicited a significantly lower number of nodules that were ineffective in nitrogen fixation (Appendix A). The swimming motility observed in the semi-solid 0.3% agar medium was substantially reduced in the case of Δ*pssA* compared to the WT both in the minimal M1 and complete 79CA media (Appendix A). These phenotypes were rescued in the Δ*pssA*(*pssA*) strain (Appendix A).

### 2.2. PssA Is an Inner Membrane Monotopic Protein with a Reentrant Helix-Break-Helix Motif

PssA was previously predicted to be an integral membrane protein [18]. Taking advantage of the substantial novelty of the content of current databases related to protein structural information, we remodeled the PssA secondary structure and its membrane topology (Figure 2A,B). The predicted topological models ambiguously assumed that PssA may be a bi- or polytopic membrane protein with both N- and C-termini facing the cytoplasm or periplasm, or with termini located on opposite sides of the membrane (Figure 2A,B). To validate these models, we employed the gene fusion approach with a dual reporter vector equipped with *phoA-lacZα* reporter sequences. The functionality of the dual reporter system was tested on two fragments of the polytopic PssT glycosyltransferase, serving a Wzy polymerase function in the EPS transport system in RtTA1. The topology of this protein was previously studied using fusions with *phoA* and *lacZ* genes, however, encoded in separate vectors [35]. In this study, two constructs were prepared encoding fusion proteins with two PssT fragments differing in the subcellular location of their C-termini (Figure 2D): T201 fusion containing a 201-aa-long fragment with the C-terminus located in the periplasm and T243 fusion containing a 243-aa-fragment with the C-terminus located in the cytoplasm [35]. The biochemical data obtained for the PssT fusions confirmed the robustness of the reporter system (Figure 2E, Table 1). The fusion protein with a 201-aa-long fragment of PssT showed high activity of alkaline phosphatase and no activity of β-galactosidase. The T243 fusion protein behaved in exactly the opposite way, exhibiting negligible activity of alkaline phosphatase and high activity of β-galactosidase (Table 1). Three fusion plasmids were constructed for the PssA protein: A85 with a fusion site in the middle of the predicted transmembrane helix, A180 with a fusion site between the third and fourth predicted α-helices, and A263 containing the whole PssA polypeptide. Since the fusion site localization in a transmembrane segment can give a combination of activities of PhoA and LacZ, the normalized activity ratio (NAR) was calculated for all the PssA and PssT fusions as in Islam et al. [36] (Table 1).

The NAR values calculated for PssT truncations supported the previously established model [35]. As indicated by the NAR values calculated for the PssA protein, A85 has a membrane location, but A180 and A263 may also interact with the membrane, most probably through amphipathic α-helices (Table 1). Considering that β-strands are predicted to form a sheet accessible in the cytoplasm for the UDP-sugar donor binding, the bitopic model framed in Figure 2B was further modified to comprise all the biochemical and computational data. PssA homology modeling revealed that the protein has a unique structural homolog, i.e., PglC, representing a monotopic protein initiating the N-linked protein glycosylation pathway in *Campylobacter* [23,37]. PglC is currently the only example of a well-characterized monotopic protein penetrating the membrane via the reentrant helix-break-helix motif immersed in the cytoplasmic leaflet of the inner membrane [23]. The MSA of the first predicted transmembrane helix of PssA with reentrant helices in the characterized PglC proteins, analyses of the propensity of certain residues to localize in exposed or buried regions, and structural modeling of this fragment of the polypeptide (Figure 2G) revealed that the predicted transmembrane helix in PssA may indeed have a helix-break-helix structure, with a “break” caused by the conserved Pro(86) residue. Like in PglC, where the positively charged Lys-Arg motif favors cytoplasmic localization of the N-terminus [23], Lys-Arg(69–70) in PssA may serve a docking function at the membrane interface (Figure 2G).

In summary, the topology predictions of PssA using multiple algorithms suggested bi- or polytopic transmembrane topology. However, the biochemical analysis of PssA-PhoALacZ fusions supported by homology modeling of the whole PssA, structural modeling, and residue conservation analyses of the predicted membrane helix supported the novel model. PssA would be a monotopic PGT with a reentrant membrane helix rather than a membrane-spanning helix (TMH) (Figure 2B).

To further substantiate the integral membrane character of PssA, a study of its subcellular localization was carried out in both the heterologous and homologous systems. For this purpose, the non-inclusion fraction obtained from *E. coli* BL21(DE3) carrying the pET30-pssA plasmid was subjected to further fractionation. The presence of the recombinant His_6_-PssA was checked by Western blotting with anti-His_6_ antibodies. An analogous experiment was also performed for the Δ*pssA* strain carrying copies of the *pssA* gene equipped with a 3′-terminal histidine tag on the medium copy vector pBBR1MCS-2 [Δ*pssA*(*pssA*his)]. The result clearly showed that, both in *E. coli* and *Rhizobium*, the PssA protein was located in a fraction of integral membrane proteins (Figure 3). Additional evidence for this strong connection with the membrane came from the expression and purification study: recombinant His_6_-PssA used in assays for donor specificity was efficiently recovered from membranes only in the presence of detergents.

### 2.3. PssA Is a Phosphoglycosyltransferase Specific for UDP-Glucose

Previously available data suggested specificity of PssA PGT for UDP-glucose [38]. However, the tests did not comprise other sugars and no documentation of this experiment was provided. Since the protein was repeatedly mentioned as an important enzyme with possible regulatory functions at the crossroads of different pathways, we intended to determine whether PssA was specific also for other UDP-sugars, namely UDP-glucuronic acid and UDP-galactose. To this end, we employed a method of rapid screening of GT donor specificity [39] using the UMP/CMP-Glo^TM^ Glycosyltransferase Assay (Promega). The assay was performed for UDP-Glc, UDP-GlcA, and UDP-Gal, and the results were represented as a ratio of the UMP measured in reactions containing recombinant His_6_-PssA relative to the control reactions with no enzyme included (Figure 4). The data obtained clearly showed PssA preference for UDP-glucose (Figure 4), supporting previously published data and confirming that the methodology of protein preparation proposed in this work retains its enzymatic activity.

### 2.4. PssA Weakly Interacts with GTs Involved in the EPS Octasaccharide Synthesis

Given the importance of PssA as a priming phosphoglycosyl transferase, we screened for possible PssA interactions with other glycosyltransferases and components of the polymerization and transport system of EPS using the bacterial two-hybrid system (BTH). Genes encoding glycosyltransferases and PssP, PssT, PssL, and PssP2 proteins were previously cloned into BTH vectors [33,40,41]. The co-transformations of the reporter strain *E. coli* DHM1, screening on the indicative media, and measurement of β-galactosidase activity revealed an interaction of PssA with one GT involved in the octasaccharide backbone synthesis (PssC) and three GTs probably involved in the side chain synthesis (PssF, PssI, PssJ) (Figure 5). Interestingly, these few hetero-interactions were observed only for PssA located at the C-terminus of the fusions, irrespective of the plasmid type (pUT or pKT derivatives), suggesting that the block of the PssA N-terminus may not affect folding and functionality of the protein.

### 2.5. The Transcriptome of the ΔpssA Mutant Reveals Changes in the Expression of Genes Not Directly Related to EPS Biosynthesis

The transcriptomes were sequenced and quantified as described in the Section 3. For individual samples, from 16,019,200 to 22,418,500 high-quality pair-end reads were obtained (Appendix A). Among a total number of 7028 CDSs annotated in the RtTA1 genome, 6982 genes were represented in this RNA-seq study (Appendix A). During the growth of Δ*pssA* in the rich 79CA medium, 79 genes (1.1% of the total gene number) were found to be significantly differentially expressed. Among those DEGs, upregulated genes predominated: 49 genes vs. 30 which were downregulated (Appendix A).

The DEGs were not evenly distributed through all RtTA1 replicons and most of them (47) were found in the chromosome while all extrachromosomal replicons in total encountered for 32 DEGs (pRleTA1d—11, pRleTA1c—9, pRleTA1b—5, pRleTA1a—7). The normalization of the number of DEGs, according to the number of CDSs harbored by each replicon, revealed that the symbiotic plasmid pRleTA1a was a replicon with the higher representation of differentially expressed genes. Similarly, the rate of differential gene expression (DEG rate) was not even for individual replicons: the highest median log_2_fold change value for the upregulated genes was found for the chromosome, while the strongest level of downregulation was observed for the pRleTA1d encoded genes (Appendix A). 

With respect to the individual DEGs, genes showing the strongest (more than 8-fold, log_2_fold change >3) downregulation of expression comprised the cluster of seven genes of pRleTA1d. The cluster encompassed RLTA1_RS27740 (carbohydrate ABC transporter substrate-binding protein), RLTA1_RS27745 (hypothetical protein), RLTA1_RS27750 and RLTA1_RS27755 (both encoding putative glycoside hydrolase family proteins), RLTA1_RS27760 (sn-glycerol-3-phosphate ABC transporter ATP-binding protein UgpC), RLTA1_RS27765 (sugar ABC transporter permease), and RLTA1_RS27770 (carbohydrate ABC transporter permease). The strongest induction of expression (more than 4-fold, log_2_fold change >2) was noticed for chromosomally encoded RLTA1_RS10060 (ABC transporter ATP-binding protein) and pRleTA1a located RLTA1_RS34375 (similar to chaperone proteins of the Hsp20/alpha crystallin family) (Appendix A). Importantly, DEGs of Δ*pssA* detected in this study comprised a totally different set of genes than the ones previously included in the putative stimulon of *pssA*, described based on proteomic analysis by Guerreiro et al. [27] (Appendix A). The low correlation between the transcriptome and the proteome was previously reported and seems to support the view that post-transcriptional processes play a major role in the adaptation of bacteria to growth conditions. It was found that the alphaproteobacterium *Rhodobacter sphaeroides* responded to sudden changes in environmental conditions by radical and rapid reprogramming of the transcriptome during the first 90 min, while modest proteome changes were observed [42]. However, in response to gradually deteriorating conditions, the transcriptome remained mostly at a steady state while the bacterium continued to adjust its proteome. Even long after the population has entered the stationary phase, the cells were still actively adjusting their proteomes. Considering the transcriptome reprogramming, the upregulation of global transcriptional regulator genes, like those of the LacI or GntR family (RLTA1_RS10065, RLTA1_RS12985, COG K), observed in the Δ*pssA* mutant seems reasonable.

Importantly, among the DEGs, no genes directly assigned to biosynthetic pathways of surface polysaccharides or peptidoglycan were found in RtTA1. The *pss* genes of the Pss-I region were of special interest, and the RNA-seq data analyses confirmed that, despite the block of EPS biosynthesis in the Δ*pssA* mutant at the initial step, there was no silencing of transcription of other *pss* genes. Alternatively, the block in the EPS biosynthetic pathway at the *pssA* step, potentially leading to an increase in the cellular pool of UDP-glucose, is likely to be filtered through to other pathways, and thus the absence of EPS may indirectly affect the levels of expression of proteins involved in these pathways, as described below.

### 2.6. Assignment of ΔpssA Mutant DEGs to COG Categories and KEGG Pathways

It was possible to assign COG categories to almost all the DEGs in Δ*pssA*. The DEGs were distributed into 14 of the 26 functional categories (Appendix A). The proportion of up- and downregulated DEGs in each COG category is summarized in Figure 6. Among the overrepresented COG categories assigned to the DEGs of the Δ*pssA* mutant were those comprising unknown function (COG S), inorganic ion transport and metabolism (COG P), amino acid transport and metabolism (COG E), and carbohydrate transport and metabolism (COG G). These categories were followed by energy production and conversion (COG C) and signal transduction mechanisms (COG T) (Figure 6). The functional COG categories of Δ*pssA* DEGs with the highest number of upregulated genes were COGs S, T, E, C, and P, while downregulated genes predominated in COGs G, H, and I (Figure 6). 

The assignment of Δ*pssA* DEGs to the KEGG pathways revealed bias towards protein families involved in signaling and general cellular processes (Appendix A) comprising pathways related to the function of ABC transporters, two-component systems (TCS) of signal transduction, biosynthesis of secondary metabolites, quorum sensing, and oxidative phosphorylation. The distribution of DEGs of the mutant in the KEGG pathways generally stayed in good agreement with the COG analysis.

### 2.7. Numerous Upregulated Genes of the ΔpssA Mutant Are Related to Amino Acid and Carbohydrate Transport and Metabolism as Well as Energy Conversion (COGs E, G, and C)

In the COG E category (amino acid transport and metabolism), DEGs encoding putative ornithine cyclodeaminase (RLTA1_RS13975), periplasmic component of ABC-type branched-chain amino acid transporter (RLTA1_RS33005), and protein belonging to the class III pyridoxal-phosphate-dependent aminotransferase family (RLTA1_RS20755) were found to be induced. The branched-chain amino acid permease (Bra)-related pathway was shown previously to be one of the dominating amino acid uptake mechanisms in *R. leguminosarum*. Due to the broad solute specificity of these ABC transporters, it may be the second general amino acid permease of *R. leguminosarum* [43]. Activation of genes related to amino acid metabolism was frequently observed in nitrogen-limited conditions [44]. In turn, pyridoxal 5′-phosphate (PLP) functions as a coenzyme in many enzymatic processes of crucial cellular metabolic pathways, including decarboxylation, deamination, transamination, racemization, and others [45]. The putative type III PLP-dependent aminotransferases comprise a class of enzymes involved in the transfer of an amino group to an acceptor, usually 2-oxo acid e.g., alpha-ketoglutarate, which is one of the key intermediates in the TCA cycle. Alpha-ketoglutarate plays an important role in the connection of carbon and nitrogen metabolisms [46]. Moreover, in the mutant strain, the RLTA1_RS32075 gene (COG S) coding for a hypothetical HutD family protein was upregulated. A study on *Pseudomonas fluorescens* showed that, in the histidine utilization (*hut*) *locus* conferring the ability of these bacteria to utilize histidine as sole carbon and nitrogen sources, HutD governs the level of *hut* operon transcription [47].

In the Δ*pssA* mutant, upregulation of several genes of COG C (energy production and conversion) was observed. Among them, there were genes encoding putative proteins of the aldehyde dehydrogenase family (RLTA1_RS19735) and phosphoenolpyruvate carboxykinase (RLTA1_RS22670). Aldehyde dehydrogenases (ALDHs) metabolize endogenous and exogenous aldehydes and thereby mitigate various stresses in prokaryotic and eukaryotic organisms. Up-regulation of ALDHs significantly contributes to the management of stress response in bacteria (both environmental and chemical) [48]. Phosphoenolpyruvate carboxykinase is involved in the gluconeogenesis in other bacteria by catalyzing the conversion of oxaloacetate (OAA) to phosphoenolpyruvate (PEP) through direct phosphoryl transfer between the nucleoside triphosphate and OAA. Moreover, three clustered genes (RLTA1_RS29620 (*cydB*), RLTA1_RS29625, and RLTA1_RS29630) assigned to COG C and RLTA1_RS29615 (*cydX*) assigned to COG S, all located in pRleTA1c and encoding the subunits of cytochrome bd ubiquinol oxidase, were found to be upregulated in the Δ*pssA* mutant. The genes were attributed to the KEGG pathway of the TCS-based signal transduction mechanism leading to the activation of cytochrome genes engaged in oxidative phosphorylation (Appendix A). Mutant cells deficient in EPS are devoid of a protective surface layer and thus may be more prone to stress during growth even in a rich medium. Increased expression of cytochromes may be a response to oxidative stress [49].

### 2.8. Activation of the Signal Transduction Mechanism (COG T) Comprises an Important Element of the ΔpssA Mutant Transcriptomic Response

COG T (signal transduction mechanisms) was a category containing exclusively upregulated genes. The upregulated genes encoded, among others, two putative proteins with similarity to PAS domain-containing sensor histidine kinases (RLTA1_RS05790 and RLTA1_RS34355) and putative response regulator CheY3 (RLTA1_RS18655); the latter protein is related to bacterial chemotaxis. The signal transduction pathways, referred to as two-component systems, are usually structured around two conserved proteins: a membrane sensor histidine protein kinase providing phosphoryl groups for response regulator proteins (transcription factor), which in turn function as molecular switches that control diverse effector activities. TCSs play a dominant role in bacterial signaling e.g., in motility and chemotaxis. In model *E. coli*, the core of the chemotaxis pathway comprises the phosphorylation of the response regulator CheY by histidine kinase CheA in response to signal transduction from the chemoreceptor proteins. Most rhizobia encode multiple chemotaxis systems and diverse chemoreceptors, e.g., *R. leguminosarum* bv. *viciae* 3841 encodes 26 such proteins. However, the main chemotactic systems of rhizobia have a similar core to that of *E. coli* [50]. 

Among the DEGs of COG T in the Δ*pssA* mutant, the *mcpV* gene (RLTA1_RS13105) encoding a putative methyl-accepting chemotaxis protein was also found to be upregulated. Such integral membrane proteins undergo reversible methylation during the adaptation of bacterial cells to environmental attractants and repellents [51]. The KEGG pathway analyses mapped the putative proteins and the response regulator described above as components of the TCS system related to bacterial chemotaxis (Appendix A). It is known that chemotaxis serves as a means of cell-to-cell communication and cell recruitment in stress conditions. Chemotaxis, in addition to the other ecological roles, is a foraging strategy by which bacteria enhance their uptake of nutrients and energy. It is also considered as protective and adaptive activity comprising a major survival strategy for bacteria [52].

In rhizobia, chemotaxis, motility, and exopolysaccharide production are linked by global regulatory pathways regulated by quorum sensing (QS) mechanisms. In *S. meliloti*, there is an inverse correlation between EPS production and motility. Distinct but related regulatory systems that enhance EPS production (MucR, ExoR/ExoS/ChvI, and ExpR/Sin) also suppress motility [25,53]. As indicated in this study, despite the lack of EPS and the upregulation of some genes related to chemotaxis, the swimming motility of the Δ*pssA* mutant observed in semi-solid agar medium was substantially reduced in comparison with that of the WT both in minimal M1 and complete 79CA media. However, passive strategies of rhizobial chemotaxis and motility have also been described. They are driven by the expansion of a growing culture, with the bacteria either producing substances to reduce friction and enable mass movement (sliding) or producing an aggregate capsule from which cells are ejected (darting) [50]. It can be speculated that bacteria lacking EPS are devoid of the passive capability of movement, concomitantly sensing some changes in culture growth conditions and activating the transcription of the respective chemotaxis and motility-related genes described above.

In COG T, RLTA1_RS14840 (*cyaJ*), the putative adenylate cyclase gene was also upregulated. Different classes of adenylate cyclases are related to various bacterial cellular pathways, as they are responsible for the synthesis of cAMP. It serves as an internal signal to activate the expression of genes for import and metabolism of sugars other than glucose. In *S. meliloti*, overexpression of *cyaJ* led to differential expression of genes related to EPS biosynthesis, synthesis of flagella, chemotaxis, and respiration [54]. Another transcriptomic study of *S. meliloti* provided evidence that the growth in Yeast Mannitol broth (YM), which is rich in carbon (due to the high levels of mannitol) and limited in nitrogen (due to the unbalanced carbon/nitrogen ratio), influenced the expression of partially overlapping genes for amino acid uptake and metabolism and *cyaJ* [44]. Many of these genes (except for EPS biosynthesis genes) were also found to be upregulated in the Δ*pssA* mutant grown in the 79CA medium, which is also unbalanced with respect to the carbon/nitrogen ratio. It has been established that *S. meliloti* accumulates substantial amounts of polyhydroxybutyrate (PHB) during growth in the YM medium. The regulatory mechanisms governing PHB accumulation are not fully clear, but it is known that the PHB cycle is linked to the regulation of other carbon storage mechanisms (exopolysaccharide and glycogen production) as well as motility and symbiosis. It can be speculated that the Δ*pssA* mutant, being unable to produce EPS during growth in the carbon-rich 79CA medium, displays disturbances in the processes of carbon storage, and engagement of CyaJ in these processes is plausible. These observations seem to be in agreement with the observed predominance of downregulated DEGs of Δ*pssA* assigned to COG G (carbohydrate transport and metabolism) (Figure 6).

Despite the above, the transcription of some genes encoding putative sugar transporters was activated (Figure 6), namely the chromosomally clustered RLTA1_RS07395, RLTA1_RS07400, and RLTA1_RS07405. They encode putative permease components of the ribose/xylose/arabinose/galactoside ABC-type transport system. According to the KEGG pathway mapping, upregulated DEGs for sugar transporters may be also engaged in the uptake of cellobiose, chitobiose, or xylitol. Differential expression (both up- and downregulation) of numerous sugar transporter genes may be one of the mechanisms to keep the balance related to carbon metabolism in cells lacking EPS. In this context, the upregulation of the RLTA1_RS34270 gene encoding hypothetical universal stress protein A family (also COG T) seems legitimate. The superfamily of universal stress proteins (USP) comprises conserved proteins encoded in the genomes of bacteria, archaea, fungi, protozoa, and plants, and their primary function is to protect the organism against environmental stress [55]. The expression of these proteins is induced by a plethora of environmental stressors: in *E. coli* the level of UspA was greatly increased during growth inhibition caused by the exhaustion of any of the variety of nutrients (carbon, nitrogen, phosphate, sulfate, amino acid) or by the presence of toxic agents, including heavy metals, oxidants, acids, and antibiotics [56,57].

### 2.9. Diverse Types of Genes Related to Stress Response Are Activated in the ΔpssA Mutant

The RLTA1_RS12190 gene of COG P (inorganic ion transport and metabolism) annotated as coding for putative ATP-binding protein UgpC, i.e., a component of the sn-glycerol-3-phosphate ABC transporter, was upregulated in the Δ*pssA* mutant. Byproducts of the glycerophospholipid metabolic pathway, such as sn-glycerol-3-phosphate, are imported for the biosynthesis of phospholipids by an ATP-binding cassette (ABC) transporter known as UgpABCE [58]. Activation of gene expression related to the synthesis of a major component of biological membranes may indicate that the mutant struggles with membrane integrity during growth as a result of the absence of the PssA membrane protein. The chromosomally encoded cluster of three genes RLTA1_RS10050, RLTA1_RS10055, and RLTA1_RS10060 (COG P) coding for putative permease components of ABC-type dipeptide/oligopeptide/nickel transport systems was upregulated in the Δ*pssA* mutant. Dpp transporters are responsible for transporting mainly dipeptides but also tripeptides into cells. They are exploited by bacteria, having limited ability to use carbohydrates as a carbon source, relying on exogenous amino acids and peptides instead [59]. 

In the COG S (unknown function), the RLTA1_RS09750 and RLTA1_RS34450 genes (located on the chromosome and pRleTA1a, respectively), coding for putative proteins with the EF-hand domain found in calcium-binding proteins, were upregulated. Calcium regulates a variety of regulatory processes in bacteria, such as motility, chemotaxis, cell division, and differentiation [60]. A calmodulin-like protein, termed calsymin, containing two predicted EF-hand Ca^2+^-binding motifs and having a confirmed calcium-binding ability was identified in *R. etli*. This protein was exclusively expressed and secreted by *R. etli* during colonization and infection of the host. A mutation of the *casA* gene coding for calsymin affected bacteroid development during symbiosis and symbiotic nitrogen fixation [61].

Moreover, another two genes of the COG S category, namely RLTA1_RS34350 and RLTA1_RS34380, coding for hypothetical proteins with the BON domain (bacterial OsmY and nodulation) were upregulated in the Δ*pssA* mutant. OsmY of *E. coli* is an outer membrane or periplasmic protein expressed in response to a variety of stress conditions, in particular osmotic shock. The architecture of BON domains suggests their contact with phospholipid interfaces surrounding the periplasmic space in response to deformations in the plasma membrane [62]. The upregulation of these genes may be related to the slightly increased resistance of the Δ*pssA* mutant to NaCl excess.

### 2.10. Downregulated Genes of the ΔpssA Mutant Are Linked to the Transport and Metabolism of Coenzymes and Lipids as Well as Translation (COG H, I, J)

Three categories, COG H and I (coenzyme and lipid transport and metabolism), and COG J (translation, ribosome structure, and biogenesis), comprised solely genes downregulated in the Δ*pssA* mutant. Genes encoding proteins with similarity to riboflavin and glutathione synthases (GSH) and genes coding for L-aspartate oxidase (NadB), i.e., an enzyme involved mainly in de novo NAD biosynthesis (also in alanine, aspartate, and glutamate metabolism), were found in GOG H. Riboflavin is the precursor of flavin mononucleotide (FMN) and flavin adenine dinucleotide (FAD). Both are typical cofactors of flavoproteins essential in multiple cellular processes, including energy production, redox reactions, light emission, biosynthesis, and DNA repair. Riboflavin availability can influence the onset of rhizobial symbiotic interactions [63].

Downregulation of expression was observed in the case of two genes of the COG J category, designated as *rpmH* and *rpsJ,* and coding for putative structural protein components of large and small ribosomal subunits, potentially involved in the binding of tRNA to ribosomes. Ribosome hibernation is a prominent molecular strategy to modulate protein synthesis during various stresses and operates in both prokaryotic and eukaryotic cells [64].

## 3. Materials and Methods

### 3.1. Bacterial Strains and Standard Culture Conditions

Bacterial strains used in this work were listed in Appendix A. *E. coli* strains were grown in lysogeny broth (LB) medium at 37 °C [65] and *R. leguminosarum* bv. *trifolii* strains were grown in 79CA with 1% mannitol [66]. *E. coli* DHM1 strain was grown at 30 °C. Antibiotics were used at the following final concentrations: 100 μg/mL ampicillin, 40 μg/mL kanamycin, 5 (*E. coli*) or 10 μg/mL (*Rhizobium*) gentamicin, 10 μg/mL tetracycline, and 40 μg/mL rifampin.

### 3.2. Bioinformatic Analyses

Sequence data were analyzed with Lasergene analysis software ver. 14 (DNASTAR, Inc., Madison, WI, USA). PssA membrane topology was predicted with The Consensus Constrained TOPology Prediction program, CCTOP [67]. Modeling of the PssA structure was done with Phyre2 [68].

### 3.3. DNA Techniques

Standard protocols for DNA isolation, PCR, molecular cloning, and transformation were used [69]. Plasmids and primers used in this work were listed in Appendix A, respectively. PCR was performed with high-fidelity Platinum SuperFi II DNA Polymerase (Thermo Fisher Scientific, Waltham, MA, USA). DNA sequencing of plasmid constructs prepared in this work was performed in Genomed (Warsaw, Poland).

### 3.4. Construction of the RtTA1 ΔpssA Mutant and PssA Complemented Strains

RtTA1 mutant deleted for *pssA* was generated using the pCM351 allelic exchange vector [32]. The regions immediately flanking *pssA* were amplified by PCR using RtTA1 genomic DNA as a template. The purified 645 bp PCR product for the *pssA* downstream region was cloned into ApaI–SacI sites of pCM351 to produce pCGpssA-D. Subsequently, the purified 584 bp PCR product for the *pssA* upstream region was introduced between KpnI–NotI sites of pCGpssA-D, resulting in pCGpssA-UD. pCGpssA-UD was then transferred to RtTA1 by biparental conjugation from *E. coli* S17-1 donor strain [70]. Bacterial mating experiments were performed as described by Reeve et al. [71]. Gentamicin-resistant transconjugants obtained on TY medium [72] containing rifampicin and gentamicin were subsequently screened for tetracycline sensitivity to identify potential null mutants. The frequency of the double-crossover event was 7.31%. One such Δ*pssA*::Gm^R^ mutant, called Δ*pssA*(Gm^R^), was chosen for further study. Analytical PCRs confirmed the successful allelic exchange. To remove the gentamicin resistance cassette, the plasmid pCM157 was introduced into Δ*pssA*(Gm^R^) by electrotransformation, as described by Garg et al. [73]. Tetracycline-resistant transformants were streaked for purity by two passages to obtain a strain called Δ*pssA*[pCM157], which produced only gentamicin-sensitive colonies. Next, pCM157 was cured from Δ*pssA*[pCM157] by five consecutive transfers on a nonselective medium to obtain the Δ*pssA* mutant strain. Analytical PCRs were performed to confirm the successful deletion of the gentamicin resistance cassette. The sequencing of PCR-amplified product indicated expected recombination between *loxP* sites.

For the construction of the pBKpssA-C plasmid used in complementation analyses, a 1739 bp PCR fragment comprising *pssA* was cloned between KpnI–XbaI sites of pBBR1-MCS2. To construct a C-terminally His6-tagged version of PssA, a 1387-bp fragment comprising *pssA* gene equipped with in-frame 3′-terminal His6-tag coding sequence was cloned into KpnI–BamHI sites of pBBR1-MCS2, resulting in pBKpssA-His6. The correctness of the constructed vectors was confirmed by sequencing. Obtained plasmids were transferred into Δ*pssA* mutant through electrotransformation, resulting in Δ*pssA*(*pssA*) and Δ*pssA*(*pssA*his) strains, respectively.

### 3.5. Phenotypic Analyses

Symbiotic performance tests, analyses of sensitivity to SDS, DOC, ethanol, and NaCl, as well as swimming motility tests were performed as described in Marczak et al. [33]. Analyses of exopolysaccharides and lipopolysaccharides produced by the mutant and the complemented strain were performed as described previously [33,34]. Briefly, the amount of exopolysaccharide secreted into the medium during cultivation in 79CA with 0.5% glycerol was determined on EPSs precipitated with 3 volumes of pure 95% ethanol from the cell-free post-culture supernatant. The sugar content was determined calorimetrically according to Dubois et al. [74] and calculated in glucose equivalents. The proportions of the Glc, GlcA, and Gal components, i.e., the glycosyl composition of EPS, were determined through the analysis of sugar derivatives, alditol acetates, by GLC-MS as described previously [41]. Lipopolysaccharides were isolated by whole-cell microextraction, analyzed, and visualized as described by Apicella [75] and Tsai & Frasch [76].

### 3.6. Construction of PhoA-LacZ Translational Fusions

Full-length and fragments of *pssA* and *pssT* genes were PCR amplified from *R. leguminosarum* bv*. trifolii* TA1 genomic DNA and cloned in *phoAlacZ*α reporter vector pPLE01 [36] using *E. coli* DH5α as an α-complementation host strain. Plasmid inserts were sequenced at Genomed (Warsaw, Poland). Color screening for β-galactosidase and alkaline phosphatase activity was performed on plates supplemented with ampicillin, IPTG, and 5-bromo-4-chloro-3-indolyl phosphate (Roche, Basel, Switzerland) or 6-chloro-3-indolyl-β-D-galactoside (Red-Gal) (Glentham Life Sciences, Corsham, UK).

### 3.7. Reporter Enzymes Quantitation

Quantitation of β-galactosidase activity was performed as described previously [33]. To quantify alkaline phosphatase activity, a protocol by Manoil [77] was followed, with minor modifications. The exponential phase cultures were cooled and the OD_600_ was measured before starting the assay. 1 mL of culture was centrifuged for 5 min, 4 °C, 4000 RCF, and the bacterial pellet was washed with 1 mL of 10 mM Tris, 10 mM MgSO_4_, pH 8.0, and centrifuged. The resulting pellet was resuspended in 1 mL of 10 mM Tris, pH 8.0, and 100 μL of the suspension was transferred to new tubes containing 900 μL of 1 M Tris, 0.1 M ZnCl_2_, pH 8.0. 50 μL of 0.1% SDS, and 50 μL of chloroform were added to these diluted suspensions, followed by shaking at 37 °C for 15 min. Samples were then transferred to 37 °C, and 100 μL of substrate solution (4 mg/mL p-nitrophenylphosphate in 1 M Tris, pH 8.0) was added. The reaction was stopped after 30 min by transferring the tubes to an ice-water bath and adding 120 μL of 0.5 M EDTA, 1 M KH_2_PO_4_ buffer. The contents were vortexed, centrifuged, and the clear supernatant was transferred to a 96-well plate for OD_420_ measurement. The alkaline phosphatase activity was calculated as follows: A = (OD_420_ × 1000)/[time (min) × OD_600_ × culture volume (mL)].

### 3.8. Overexpression and Purification of Recombinant PssA Protein

*E. coli* C41(DE3) strain carrying the recombinant pET30-pssA plasmid was grown at 37 °C in LB medium supplemented with kanamycin and ethanol at a final concentration of 2% (*v*/*v*), in 1 L culture, until OD_600_ of 0.7 was reached. IPTG was added to a final concentration of 0.1 mM and incubation continued for 18 h at 28 °C with shaking. Cells were harvested by centrifugation (5000 RCF, 10 min, 4 °C), then resuspended in 50 mL of lysis buffer (50 mM NaH_2_PO_4_, 300 mM NaCl, pH 7.3) with lysozyme (1 mg/mL; Sigma-Aldrich), protease inhibitor cocktail (10% of final volume; Sigma Aldrich) and Viscolase (0,025 U/μL; A&A Biotechnology). After one hour of incubation with gentle mixing on ice, the cell suspension was subjected to disintegration with a French Pressure Cell Press (18 000 psi) Thermo Fisher Scientific, Waltham, MA, USA). The resulting cell lysate was centrifuged to remove larger cell fragments (4000 RCF, 5 min, 4 °C). The clarified lysate was used to purify the recombinant PssA enzyme. Clarified lysate proteins were solubilized with Tween-20 detergent at a final concentration of 0.1% during an overnight incubation at 4 °C on a laboratory cradle. The lysate was then centrifuged (10,000 RCF, 10 min, 4 °C) to remove inclusion bodies. The supernatant was mixed with nickel resin (INDIGO Ni-Agarose, Cube Biotech) previously calibrated with lysis buffer and incubated for 2 h with gentle mixing on ice. Buffers used for washes contained 10–20 mM imidazole, while those for protein elution—500 mM imidazole. The bed protein purification step was performed twice. In the case of the second purification, fractions eluted from the column were used after subjecting them to the dialysis process using Slide-A-Lyzer ™ dialysis cassettes (Thermo Fisher Scientific, Waltham, MA, USA, 20 K MWCO) according to the protocol, in a dialysis buffer composed of 50 mM NaH_2_PO_4_, 300 mM NaCl, 0.1% Tween-20, pH 7.3. After the second purification step, this dialysis process was carried out in an analogous manner. The final concentration of the enzyme was measured with the Pierce™ Coomassie Plus (Bradford) Assay Kit (Thermo Fisher Scientific, Waltham, MA, USA). The obtained protein fractions eluted from the resin were analysed by SDS-PAGE, visualized by PageBlue Staining Solution (Thermo Fisher Scientific, Waltham, MA, USA), and electroblotted onto a PVDF membrane (Merck Millipore, Burlington, MA, USA). Immunoblots were probed with anti-His6 antibodies (Thermo Fisher Scientific, Waltham, MA, USA) and secondary anti-mouse IgG conjugated with alkaline phosphatase (Roche, Basel, Switzerland).

### 3.9. Protein Localization Study

For the isolation of soluble and membrane proteins, bacterial pellets were resuspended in 50 mM Tris-HCl (pH 8.0) supplemented with protease inhibitors and lysozyme (final concentration 2 mg/mL). The disintegration of cells was performed with the French Pressure Cell Press at 18,000 psi. Fractionating centrifugation was then performed to obtain clarified lysate (4000 g), inclusion bodies and non-inclusion fraction (10,000 g), soluble and total membrane proteins (100,000 g), and integral membrane and membrane-associated proteins (100,000 g after 1 M NaCl washing).

### 3.10. UDP-Sugar Hydrolysis

The ultra-pure UDP-Glc, UDP-Gal, and UDP-GlcA as well as UMP/CMP-Glo™ Glycosyltransferase Assay Kit were purchased from Promega Corporation, Madison, WI, USA. The measurement of the donor specificity of the recombinant PssA, determined as the hydrolysis of the nucleotide sugar derivative, was carried out by incubating a given amount (0.1–5 μg) of recombinant His6-PssA in the reaction buffer: 50 mM HEPES, 100 mM NaCl, 5 mM MgCl_2_, 0.1% Tween-20 [23] with a given sugar substrate (100 μM) for 4 h at 28 °C. Control mixtures contained no enzyme. The hydrolysis reaction was stopped by adding UMP/CMP Detection Reagent^TM^ in a 1:1 ratio (25 μL:25 μL). After mixing, the plate was incubated at room temperature for 1 h. Then the level of luminescence was measured with Synergy H1 multi-detection reader (BioTek, Agilent Technologies, Santa Clara, CA, USA). The intensity of emitted light (luciferase reaction) is proportional to the amount of free UMP produced in the UDP-sugar hydrolysis reaction. Each reaction was performed in a separate well of a 96-well plate in two technical repeats.

### 3.11. Bacterial Two-Hybrid System

Plasmids encoding fusion proteins GT-CyaA or CyaA-GT, as well as plasmids encoding protein components of the Wzx/Wzy-dependent transport and polymerization system of EPS (PssT, PssP, PssL, and PssP2 proteins), were previously constructed [33,40,41] and are listed in Appendix A. Plasmids were co-transformed into *E. coli* DHM1 strain and interaction screening was performed using agar plates containing 5-bromo-4-chloro-3-indolyl-β-D-galactoside (X-Gal; 40 μg/mL), isopropyl-β-D-galactopyranoside (IPTG; 0.5 mM), ampicillin (100 μg/mL), and kanamycin (40 μg/mL). Quantitative measurement of β-galactosidase activity was performed in a microplate format as described earlier [33].

### 3.12. RNA Extraction

Cultures of *R. leguminosarum* bv. *trifolii* TA1 and Δ*pssA* mutant strains were grown in triplicate for 24 h in 79CA at 28 °C with shaking, then diluted to an OD_600_ of 0.05 in fresh 79CA medium and incubated until an OD_600_ of 0.7 was reached (≈10^9^ CFU). The cells were harvested by centrifugation at 4 °C and immediately submitted for RNA extraction with the commercial GeneMATRIX Universal RNA Purification Kit (EURx Sp. z o.o., Gdańsk, Poland) according to the manufacturers protocol. Contaminating DNA was removed with TURBO DNA-free™ Kit (Thermo Fisher Scientific, Waltham, MA, USA), under rigorous treatment protocol, with minor modifications. The quantity and quality of RNA were checked spectrophotometrically (Synergy H1 reader, Agilent Technologies, Inc., Santa Clara, CA, USA), fluorometrically (Qubit 2.0 Fluorometer with Qubit RNA High Sensitivity (HS) Assay Kit, Thermo Fisher Scientific, Waltham, MA, USA), in microcapillary electrophoresis (2100 Bioanalyzer Instrument with RNA 6000 Nano Kit, Agilent Technologies, Inc., Santa Clara, CA, USA), and in the PCR reaction. Details regarding the optimization of high-quality RNA extraction were presented in Appendix A.

### 3.13. RNA Sequencing and Data Analysis

The depletion of the ribosomal RNA and cDNA library preparation was performed with Illumina Stranded Total RNA Prep with Ribo-Zero Plus (Illumina Inc., San Diego, CA, USA). The samples were sequenced by mean NovaSeq6000 platform with 2 × 150 bp sequencing mode. The resulting raw reads files for the six transcriptomes have been deposited in the SRA of NCBI and are available via accession PRJNA894372. Poor quality reads and adapter removal was performed with Trimmomatic [78]. The quality of the sequences was evaluated with FastQC (http://www.bioinformatics.babraham.ac.uk/projects/fastqc/ (accessed on 29 March 2021)). To map the reads to the reference genome of *R. leguminosarum* bv. *trifolii* TA1 (RefSeq assembly: GCF_000430465.3), Bowtie2 software was used with default parameters [79], and non-uniquely mapped reads were filtered with SAMtools [80]. Normalized differential expression of genes (DEG) was calculated using DESeq2 [81], limma [82], NOISeq [83], edgeR [84] software packages available at Bioconductor—OpenSource Software for Bioinformatics (www.bioconductor.org (accessed on 1 January 2023)) (Appendix A). The following thresholds were applied for DEGs determination: *p*-adjusted value ≤ 0.05, log_2_fold change equal to or less than −1 or equal to or greater than 1 (1.0 ≤ log_2_FC ≥ 1.0), and CPM (Counts Per Million) ≥1 (the latest value removes from analysis of differential gene expression those genes, whose expression is very low). The selected DEGs comprised a common set for all four statistical methods used, based on the criteria established above. Functional enrichment and annotation of DEGs was performed with reCOGnizer [85], eggNOG v5.0 [86], GSEA-pro [87], and KEGG mapper [88,89].

## 4. Conclusions

The data provided in this work seem to exclude the previously hypothesized direct wide engagement of PssA in various cellular processes or even its regulatory role in rhizobial cells. It is certain that PssA is a phosphoglycosyl transferase specific for UDP-glucose and priming the synthesis of the exopolysaccharide octasaccharide subunit in *Rhizobium leguminosarum.* PssA was shown to be a membrane protein with a specific and unusual monotopic topology, described recently for only a limited number of proteins. The wide distribution of DEGs in the Δ*pssA* mutant transcriptome into various functional COG categories denotes the complexity of cell response to deletion of a single non-regulatory gene. The picture obtained in our RNAseq experiment suggests that most of the changes in the gene expression profile of Δ*pssA* can be attributed to the lack of exopolysaccharide and unbalanced carbon storage and stress protection. The absence of a single protein itself seems to have only minor consequences for bacterial physiology, e.g., changes in the membrane structure due to the absence of a membrane protein, besides the obvious blockage of EPS synthesis caused by the lack of crucial enzymatic activity. The phenomenon of the Δ*pssA* mutant supports the common concept that linking the diverse and distantly related phenotypes directly with the mutated gene may be misleading even in bacteria, as the phenotype can rather be attributable to downstream effects of a particular mutation.

## Figures and Tables

**Figure 1 ijms-24-01035-f001:**
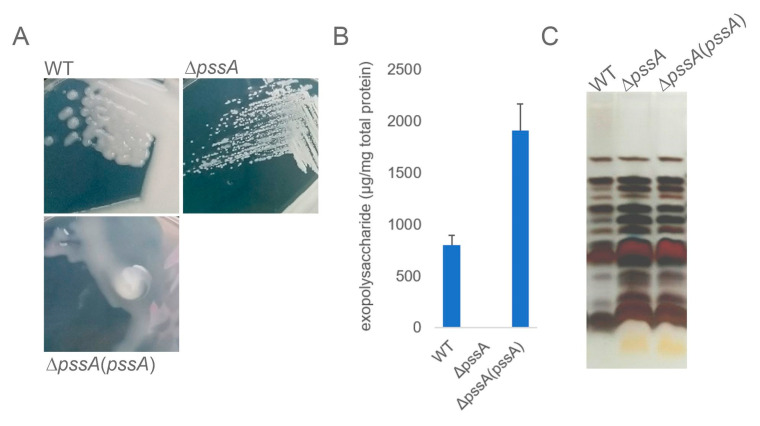
Analyses of polysaccharides produced by the Δ*pssA* mutant and its complemented derivative. (**A**) Microscopic morphology of single colonies formed on 79CA agar plates. (**B**) Result of the quantification of exopolysaccharides secreted into the liquid culture. The amount of sugar in tested samples was determined based on the calibration curve prepared for glucose and expressed as the total sugar content per amount of total bacterial protein (μg/mg). Error bars represent standard deviation. (**C**) Profiles of LPS of the tested strains separated in Tricine-PAGE electrophoresis and stained with silver.

**Figure 2 ijms-24-01035-f002:**
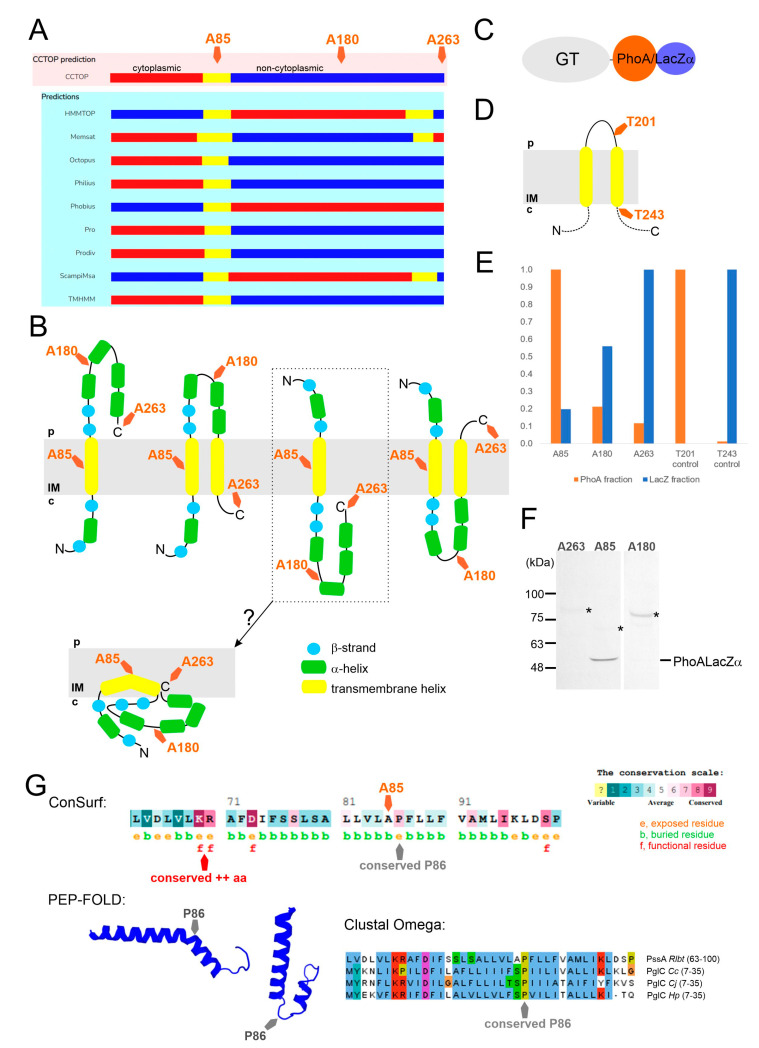
PssA phosphoglycosyl transferase topology prediction, analyses, and modelling. (**A**) Graphical presentation of CCTOP meta-tool topology predictions being the result of consensus for ten different methods. (**B**) Four possible topology models resulting from combined CCTOP and Phyre2 predictions; dotted frame surrounds the only bitopic model partially supported by the biochemical data, and further elaborated based on the analyses summarized in panel G. (**C**) Schematic representation of dual reporter translational fusions construction. (**D**) Schematic representation of PssT control periplasmic and cytoplasmic fusions construction; dotted lines indicate that the scheme presents only a fragment of this 12-TMS-protein. (**E**) Reporter enzymes activity quantification—the graph presents relative values, i.e., the percentage of PhoA and LacZ activities of the fusions regarding the maximum activity in the set for PssA and PssT alone. (**F**) Western blotting result showing the level of expression and stability of PssA fusion proteins; bands marked with asterisks represent PssA fusion proteins with predicted molecular masses: A85—64.4 kDa, A180—74.8 kDa, and A263—83.9 kDa. (**G**) Analyses of the PssA first transmembrane helix predicted by all bioinformatic tools employed: ConSurf prediction of functional/structural conservation of residues, as well as their exposure or burial; the position of Ala85 fusion site is indicated; Pro86 is predicted to be conserved, and Lys-Arg(69–70) are predicted to be highly conserved and responsible for docking of the helixat the cytoplasmic side of the membrane. PEP-FOLD structural modelling of PssA transmembrane helix predicted to form “helix-break-helix” structure characteristic of reentrant membrane helix (RMH) described for PglC *Campylobacter jejuni.* Multiple sequence alignment (MSA, Clustal Omega) of the reentrant membrane helices present in PssA and three structurally characterized monotopic phosphoglycosyltransferases: PglC *Campylobacter concisus* (*Cc*) (A7ZET4), PglC *C. jejuni* (*Cj*) (Q0P9D0), and PglC *Helicobacter pullorum* (*Hp*) (E1B268) is shown at the bottom right. Numbers in brackets represent UniProt IDs. Alignment was visualized with Jalview, and Clustal coloring scheme was applied.

**Figure 3 ijms-24-01035-f003:**
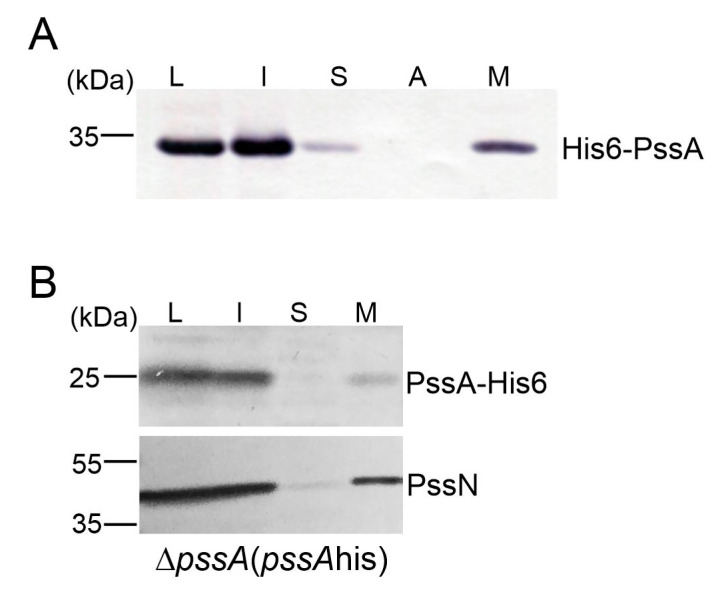
Study of the subcellular localization of PssA. Proteins from cell fractions obtained from *E. coli* BL21(DE3) carrying expression construct pET30-pssA (**A**) and Δ*pssA* strain carrying *pssA* cloned into pBBR1MCS-2 [Δ*pssA*(*pssA*his)] (**B**) were separated in SDS-PAGE and subjected to Western blotting with the anti-His_6_ (for localization of tested proteins), and anti-PssN (for verification of the effectiveness of the *Rhizobium* cells fractionation) antibodies. The experiment was repeated twice giving the same result. L—cell lysate, I, S, M—insoluble, soluble and membrane fraction of proteins, A—fraction of membrane-associated proteins (released after NaCl washing).

**Figure 4 ijms-24-01035-f004:**
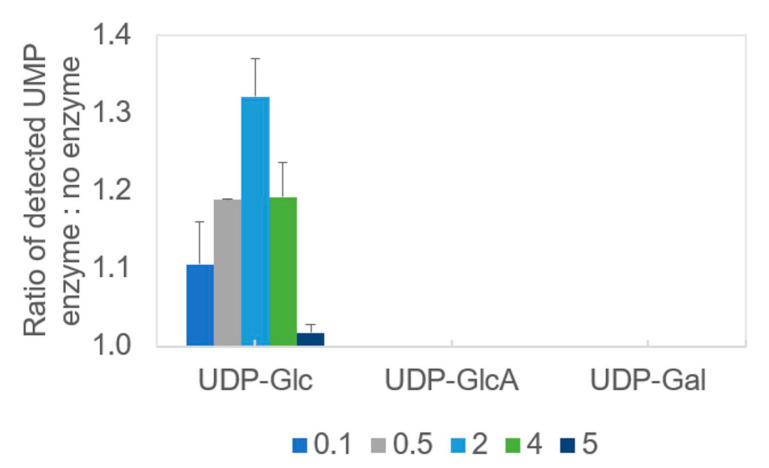
Determination of the donor specificity of recombinant PssA phosphoglycosyltransferase. The colored bars represent the averaged results of two replicates for a given protein amount indicated in the legend at the bottom (μg), for which the ratio of the UMP detected in the complete reaction to the respective negative control (no enzyme) was calculated—thus the *X*-axis intersects the *Y*-axis at 1.0. Error bars represent standard deviation.

**Figure 5 ijms-24-01035-f005:**
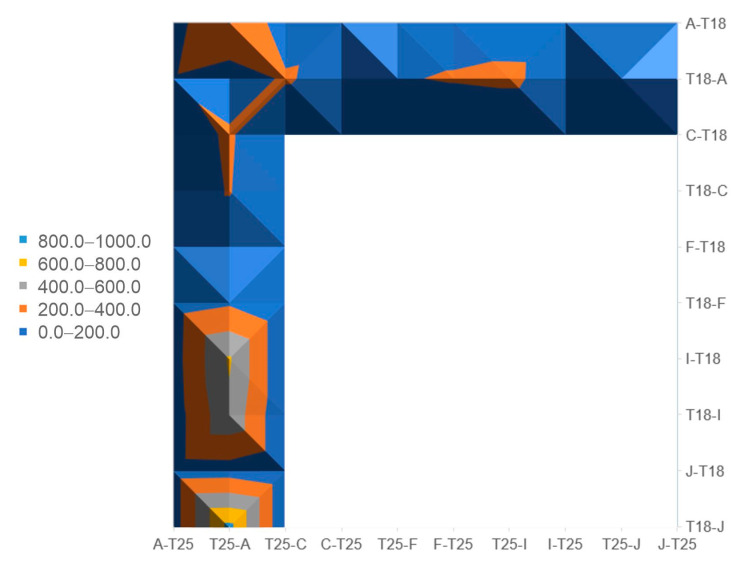
Bacterial two-hybrid screening for PssA interactions with other glycosyltransferases and components of the polymerization and export system of exopolysaccharide in *R. leguminosarum* bv. *trifolii*. The surface diagram includes only those protein-protein pairs tested, where activities twice as high as the values for the negative controls were observed at least in one of eight possible combinations of fusion proteins. β-galactosidase activities of negative controls [pUT18(pUT18C) × pKT25(pKNT25)] ranged from 90.7 ± 4.9 to 98.9 ± 2.1. Ranges of activity values in the graph refer to Miller units. Activity values ± SD for all the tested fusion proteins pairs were given in a Appendix A.

**Figure 6 ijms-24-01035-f006:**
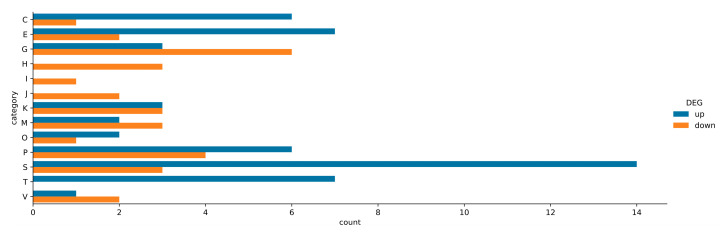
Representation of COG categories comprising DEGs of *R. leguminosarum* bv. *trifolii* Δ*pssA* mutant. COG categories: S—function unknown, C—energy production and conversion, E—amino acid transport and metabolism, G—carbohydrate transport and metabolism, H—coenzyme transport and metabolism, I—lipid transport and metabolism, J—translation, ribosomal structure and biogenesis, K—transcription, M—cell wall/membrane/envelope biogenesis, O—posttranslational modification, protein turnover, chaperones, P—inorganic ion transport and metabolism, T—signal transduction mechanisms, and V—defense mechanisms.

**Table 1 ijms-24-01035-t001:** Calculation of normalized PhoA:LacZ activity ratio (NAR) and proposed truncation sites location in a given protein.

Fusion	Avg PhoA ^1^	Avg LacZ ^1^	PhoA (%) ^2^	LacZ (%) ^2^	NAR (PhoA:LacZ) ^3^	Location ^4^
A85	1195.4	166.0	100	20	5	tm
A180	254.4	467.3	21	56	0.375	tm-c
A263	140.5	835.4	12	100	0.120	tm-c
T201	873.5	0	100	0	>100	p
T243	11.6	684.2	0.01	100	<0.01	c

^1^ Alkaline phosphatase (PhoA) and β-galactosidase (LacZ) activities of the fusions were determined as described in Section 3; values represent averages of at least four independent experiments, with two technical repeats each; activity measured for *E. coli* DH5α carrying the empty vector pPLE01 were subtracted from the presented values. ^2^ Percentage of PhoA and LacZ activities of the fusions regarding the maximum activity in the set for PssA and PssT. ^3^ Normalized PhoA:LacZ activity ratio (NAR): >10, periplasmic residue; 0.1–10, transmembrane residue; <0.1, cytoplasmic residue. ^4^ Proposed locations of the fusion points on the topological map of PssA: tm, transmembrane; p, periplasmic, and c, cytoplasmic.

## Data Availability

The files containing raw reads of the transcriptomes analyzed in this work have been deposited in the SRA of NCBI and are available via accession PRJNA894372.

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
