# Peer review of "A New Face of the Old Gene: Deletion of the PssA, Encoding Monotopic Inner Membrane Phosphoglycosyl Transferase in Rhizobium leguminosarum, Leads to Diverse Phenotypes That Could Be Attributable to Downstream Effects of the Lack of Exopolysaccharide"

_ijms, 2023, doi:10.3390/ijms24021035_

Round 1

Reviewer 1 Report

Manuscript in the current format cannot be accepted for the publication, long sentences and grammar needs to be thoroughly checked. In my opinion minor revision is required prior to the acceptance of the paper.

Minor question:

1. Why did the  ΔpssA (pssA) mutant produced 5x higher EPS than the wild type, are the gene copies and the promoters used is similar to the wild type? what author want to prove using this strain?

2. In the SDS page of  ΔpssA and  ΔpssA(pSSA) I could not understand or detect any difference, reason?

3. With interest, do author have any idea on the composition of this EPS and the concentrations it is produced?

Author Response

Manuscript in the current format cannot be accepted for the publication, long sentences and grammar needs to be thoroughly checked. In my opinion minor revision is required prior to the acceptance of the paper.

The long sentences and grammar were corrected in the revised version of the manuscript, and we hope that now it is easier to read.

Minor question:

  1. Why did the  ΔpssA (pssA) mutant produced 5x higher EPS than the wild type, are the gene copies and the promoters used is similar to the wild type? what author want to prove using this strain?

The results shown for ΔpssA(pssA) complemented strain proved the that lack of EPS production in the mutant was specifically dependent of the pssA gene deletion. This increase in EPS production due to the overexpression of pssA was previously shown by Janczarek et al. (2009), and due to the specific construction of the complementation plasmid in this work (medium copy vector, gene cloned under Plac) we suspect that overexpression of pssA could be the case. However, the intention of this complemented mutant construction was only to confirm that the phenotype was dependent on pssA deletion and not any other secondary mutations.

  1. In the SDS page of  ΔpssA and  ΔpssA(pSSA) I could not understand or detect any difference, reason?

The intention of this experiment was to verify if pssA deletion had any influence on LPS production. In fact, the presented gel shows no differences between the three presented strains: the WT, the mutant, and the complemented strain. The fact that some bands are faint in one strain and sharper in the other does not prove any structural changes.

  1. With interest, do author have any idea on the composition of this EPS and the concentrations it is produced?

The composition of exopolysaccharide produced by R. leguminosarum strain was characterized and shown to be 5:2:1 Glu:GlcA:Glc. It was similar in EPS samples of the WT, the mutant, and the complemented strain. Concentration of sugars in the wild-type exopolysaccharide was 802.1±94.5 μg sugar/mg total protein of the cells from the same sample.

Reviewer 2 Report

A comprehensive analysis of the pssA gene function and its protein structure and function have been investigated using various advanced methods. The finding of the study clearly shows that pssA is a significant monotopic inner membrane protein with a reentrant membrane helix rather than a transmembrane segment that a phosphoglycosyl transferase specific for UDP-glucose priming the synthesis of the exopolysaccharide octasaccharide subunit in R. leguminosarum. This study work can be considered after revision of the comments given below:

-Lines 129-136: A statement was made on the findings of the study at the end of the Introduction. It would be better not to give the findings of the study in the Introduction.

- The resolution of all the figures is very low, especially since the text on the figures is faint and not readable.

-Table 1 endnotes describing the superscript are all about 1, but 2 is missing, and 3 and four are not described.

-The findings stated in the Abstract are not stressed in the conclusion section as well. Therefore, it leads the reader to confusion about the significant outcomes of the study.

-Some sentences in some parts of the manuscript make it difficult to understand the meaning. It would be better to go through the text to revise such long sentences to make them easily understandable.

Author Response

A comprehensive analysis of the pssA gene function and its protein structure and function have been investigated using various advanced methods. The finding of the study clearly shows that pssA is a significant monotopic inner membrane protein with a reentrant membrane helix rather than a transmembrane segment that a phosphoglycosyl transferase specific for UDP-glucose priming the synthesis of the exopolysaccharide octasaccharide subunit in R. leguminosarum. This study work can be considered after revision of the comments given below:

-Lines 129-136: A statement was made on the findings of the study at the end of the Introduction. It would be better not to give the findings of the study in the Introduction.

This was corrected according to the Reviewer suggestion.

 -The resolution of all the figures is very low, especially since the text on the figures is faint and not readable.

Higher resolution figures were inserted to replace the former versions.

 -Table 1 endnotes describing the superscript are all about 1, but 2 is missing, and 3 and four are not described.

Table endnotes were corrected to include missing 2 – 4.

 -The findings stated in the Abstract are not stressed in the conclusion section as well. Therefore, it leads the reader to confusion about the significant outcomes of the study.

The conclusion section was revised according to the Reviewer suggestion.

-Some sentences in some parts of the manuscript make it difficult to understand the meaning. It would be better to go through the text to revise such long sentences to make them easily understandable.

The long sentences and grammar were corrected in the revised version of the manuscript, and we hope that now it is easier to read.